# Trans-Disciplinary Responses to Climate Change: Lessons from Rice-Based Systems in Asia

Jon Hellin [1,*], Jean Balié [2], Eleanor Fisher [3], Ajay Kohli [4], Melanie Connor [1], Sudhir Yadav [1], Virender Kumar [1], Timothy J. Krupnik [5], Bjoern Ole Sander [6], Joshua Cobb [7], Katherine Nelson [6], Tri Setiyono [1], Ranjitha Puskur [1], Pauline Chivenge [1] and Martin Gummert [1]

1   Sustainable Impact Platform at the International Rice Research Institute (IRRI), DAPO Box 7777, Metro Manila 1301, Philippines; m.connor@irri.org (M.C.); s.yadav@irri.org(S.Y.); virender.kumar@irri.org(V.K.); t.setiyono@irri.org(T.S.); r.puskur@irri.org(R.P.); p.chivenge@irri.org (P.C.); m.gummert@irri.org (M.G.)
2   Agri-Food Policy Platform at the International Rice Research Institute (IRRI), DAPO Box 7777, Metro Manila 1301, Philippines; j.balie@irri.org
3   School of Agriculture, Policy and Development at the University of Reading, Reading, RG6 6AR, UK; e.fisher@reading.ac.uk
4   Strategic Innovation Platform at the International Rice Research Institute (IRRI), DAPO Box 7777, Metro Manila 1301, Philippines; a.kohli@irri.org
5   Sustainable Intensification Program, International Maize and Wheat Improvement Center (CIMMYT), House 10/B. Road 53. Gulshan-2, Dhaka 1213, Bangladesh; t.krupnik@cgiar.org
6   Sustainable Impact Platform at the International Rice Research Institute (IRRI), c/o Agricultural Genetics Institute Km 2 Pham Van Dong Str., Tu Liem District, Hanoi 100000, Vietnam; b.sander@irri.org (B.O.S.); k.nelson@irri.org (K.N.)
7   Rice Breeding Platform at the International Rice Research Institute (IRRI), DAPO Box 7777, Metro Manila 1301, Philippines; j.cobb@irri.org
*   Correspondence: j.hellin@irri.org

**Abstract:** Climate change will continue to have a largely detrimental impact on the agricultural sector worldwide because of predicted rising temperatures, variable rainfall, and an increase in extreme weather events. Reduced crop yields will lead to higher food prices and increased hardship for low income populations, especially in urban areas. Action on climate change is one of the Sustainable Development Goals (SDG 13) and is linked to the Paris Climate Agreement. The research challenge posed by climate change is so complex that a trans-disciplinary response is required, one that brings together researchers, practitioners, and policy-makers in networks where the lines between "research" and "development" become deliberately blurred. Fostering such networks will require researchers, throughout the world, not only to work across disciplines but also to pursue new South–North and South–South partnerships incorporating policy-makers and practitioners. We use our diverse research experiences to describe the emergence of such networks, such as the Direct Seeded Rice Consortium (DSRC) in South and Southeast Asia, and to identify lessons on how to facilitate and strengthen the development of trans-disciplinary responses to climate change.

**Keywords:** climate change; trans-disciplinary networks; rice-based systems; South and Southeast Asia

## 1. Introduction

Climate change remains among the most potent challenges to agricultural development. It undermines productive capacity and disrupts food markets [1]. Food supply is affected through

climate variability and shocks that negatively affect productivity. Apart from the unfavorable long-term primary and secondary impacts of global climate change on agriculture and human development [2–5], a more immediate concern is the increase of extreme events such as droughts and floods [6–8]. Small-scale farmers in the tropics are particularly vulnerable because they often farm marginal land [9,10] and have limited adaptive capacity. Therefore, it is critical to build resilience of food production systems to climate change for both increased food security, poverty reduction and enhanced social equity.

The urgency of addressing these issues is underscored by modeling evidence predicting significant decline in crop yields by 2030 [11]. Climate extremes may exceed critical thresholds for agriculture; thus, effective mechanisms to reduce production risk will be needed. Given this background, actions to transform agriculture in response to climate change are critical. Moreover, in line with the Sustainable Development Goals, climate action also needs to address gender equity and socially-inclusive development [12,13].

A recent comment in *Nature Climate Change* [14] noted that the research challenges posed by climate change are so complex that a trans-disciplinary response is required, bringing together networks of researchers, practitioners, and policy-makers. Fostering such networks will require researchers not only to work across disciplines but also to pursue new South–North and South–South partnerships. The authors also added that in the area of climate change research, there has been "*a near whole-sale shift toward applied science, and a recognition that scientists must engage in messy and complex processes of policy development*" [14].

The agricultural research-for-development (AR4D) community posits that specific technologies and innovation practices will enable farmers to adapt to the adverse impacts of climate change. One set of innovations relates to climate-smart agriculture (CSA), which has emerged as an approach to transform and reorient agricultural systems to achieve food and livelihood security under climate change. CSA is a set of guiding principles for farmers to adapt to growing natural resource constraints and increasingly unpredictable weather conditions [15,16]. CSA is defined by three objectives: (i) increasing agricultural productivity to support increased incomes, food security, and development; (ii) increasing adaptive capacity and resilience to climate variability at multiple levels (from farm to nation); and (iii) decreasing greenhouse gas emissions where possible and appropriate [17].

CSA includes, but is not limited to, climate-adapted crop varieties, and crop and land management practices that enhance resource-use efficiency and reduce greenhouse gas emissions such as site-specific nutrient management, laser land leveling, resource efficient tillage and crop establishment methods, and efficient water management. Scaling or widespread farmer uptake of CSA is challenging. The traditional linear approach for technology development and transfer involves upstream research institutions that engage in scientific discovery and proof-of-concepts, which once validated, are handed over to downstream practitioners for piloting, who in turn transfer technologies and products to extension services who pass them to farmers. The focus has now shifted to the facilitation of learning and joint action in multi-stakeholder settings, often in relation to innovation systems [17]. In this context, an active and continuous reassessment of the necessary field conditions, genotypes, agronomic management practices, and enabling policies calls for a continual stream of validated upstream research products suitable for downstream adoption and adaptation.

We use our combined and diverse experiences from climate change research in South and Southeast Asia to illustrate ways to foster trans-disciplinary research teams and to pursue South–North and South–South partnerships. We do not suggest that what follows is a blueprint for fostering and sustaining trans-disciplinary responses, but rather an example of an approach that can be readily adapted to different circumstances. This introductory Section 1 is followed by Section 2 that describes the building of interdisciplinary climate change research at the International Rice Research Institute (IRRI), an international agricultural research-for-development (AR4D) organization based in the Philippines and with country offices throughout Asia. In Section 3, we describe some of the South–North and South–South trans-disciplinary partnerships that have been established in South and Southeast Asia as

part of climate change adaptation, mitigation, and transformation efforts. Finally, in Section 4, we draw lessons from our experiences.

## 2. Building Interdisciplinary Research Teams to Address Climate Change

### 2.1. Climate Change Challenges to Rice Production in South and Southeast Asia

South Asia and Southeast Asia are priority regions for climate action largely because of the predicted impact on the production of rice: the main staple in these regions [18]. Chronic and sporadic water shortages, coupled with flooding, could derail the impressive rate of economic growth in South and Southeast Asia over the last few decades, with direct and adverse negative effects on the livelihoods of farmers and other value chain actors relying on rice production. Data from the Emergency Events Database (EM-DAT; http://www.emdat.be/database), show that extreme climate events (e.g., droughts, floods, storms, and typhoons) have occurred more frequently in the last twenty years (1999–2018) than the previous two decades (1979–1998) in rice-producing countries of Southeast and South Asia.

Drought is one of the constraints to rice production in South and Southeast Asia where approximately 23 million ha of rice (20% of total rice area) are mostly in rain-fed areas [19]. Flooding is also a major threat. Although rice can thrive when its roots are subjected to flooding, it cannot survive prolonged submergence. Energy reserves are rapidly exhausted when plant tissues respond through elongation when exposed to prolonged submergence. This can cause death within a matter of days [20]. At present, ~20 million ha of rice are prone to submergence caused by flash flooding, mostly in India and Bangladesh, partly due to cyclones but also because of increased river discharge due to increased precipitation in the watersheds. The most severe recent floods that affected Bangladesh were in 1988 and 1998 when 60% of the country was submerged [21].

Another growing threat to Asia's rice-growing areas is sea level rise as a consequence of climate change [22]. River deltas in South and Southeast Asia are extremely vulnerable, a conclusion that is aggravated by recent findings that mean elevation of some deltas is much lower than previously estimated [18,23]. Asia hosts many huge river deltas where most of the population lives. The detrimental impact of sea level rise varies depending on local geography, population distribution, producers' resource endowments, and state of preparedness. Sea-level rise can lead to frequent and hazardous flooding, soil and water salinization, and coastal erosion. In some cases, the impacts may lead to a substantial loss of production capacity and habitat to the extent that agricultural-based climate change adaptation may not be sufficient. In response, people will have to migrate [24].

### 2.2. Climate Smart Agriculture (CSA)

Researchers have developed a plethora of CSA technologies and practices for Asia including drought-, submergence-, and saline-tolerant rice varieties, improved crop and land management practices, modeling approaches, and geospatial tools to assess damage from floods and droughts. CSA research brings together different people including those working on upstream genomic research, crop breeding, agronomy, social development, etc. In turn, these researchers are part of the complex impact pathway, the functioning of which ultimately determines whether CSA is developed, adopted, and adapted, and whether it leads to climate change adaptation and mitigation benefits in a transformative way. Examples of CSA for rice-based systems in Asia include the following.

- Rice varieties tolerant to certain levels of drought, flooding, salinity, and heat.
- Alternate wetting and drying (AWD) to mitigate $CO_2$ emissions and achieve water savings.
- Direct seeding of rice (DSR) as an alternative to puddled transplanted rice (PTR) to adapt to water shortages, to mitigate GHG emissions, and to address labor shortages.
- Laser land leveling and sustainable water management practices that reduce GHG emissions and increases water efficiency.
- Sustainable rice straw value chains that reduce straw burning and GHG emissions.

- Site-specific nutrient management that enhances resource-use efficiency.
- Geospatial tools to estimate rice production and assess damage from floods and droughts, providing data quickly to insurance schemes.
- Integrated pest management (IPM) and weed management (IWM) practices to manage emerging insect–pest, disease, and weed problems.

The process of putting CSA into practice does not follow a linear transfer-of-technology approach [25]. Increasingly, end-users (farmers and other value chain actors) are involved from the beginning of the research process even though not all are always adequately represented. For example, there is extensive evidence that women and other vulnerable social groups do not play a large role in influencing R4D priorities. This needs to be rectified because technologies are not "neutral" and their uptake can exacerbate social and gender inequalities [26]. However, CSA technologies can be targeted and implemented via context-specific and community-driven approaches to ensure greater gender and social equity [27].

Adaptation to, and mitigation of, the effects of climate change are accelerated and are more effective through highly interdisciplinary research collaborations [14]. However, working across disciplines is rarely considered in a broader sense and is often manifested in a vertical axis, whereby practitioners of different but affiliated disciplines collaborate. For example, collaborations between a botanist and a biophysicist; a physiologist and a breeder; a pedologist and a water management expert or an economist and a policy specialist are common. Less common are collaborations between several disciplines, and when it happens it tends to be multi-disciplinary (different disciplines working together each relying on their disciplinary knowledge) rather than interdisciplinary (bringing together different disciplines and creating a comprehensive framework beyond one disciplinary perspective).

Effective communication among researchers from different disciplines is challenging given the potential for different or conflicting interests and the fragmentary nature of scientific disciplinary language. Additionally, a growing emphasis in climate change research on applied science has left people working in upstream research feeling as though their research is increasingly disconnected from downstream application. Bringing research scientists together from different disciplines can perhaps be seen as the scientific equivalent of the Tower of Babel, in which effective communication is stymied by the absence of a common language.

Our experience to date is that the lack of effective communication amongst different disciplinary researchers arises more from a lack of forums in which researchers come together to identify commonalities than it does from the absence of a common language. Regular open and frank discussions allow for a greater appreciation of a particular discipline's key function in an impact pathway. This can encourage continuous reassessment of the necessary genotypes, agronomic management practices, livelihood analyses and enabling policies for climate change adaptation, mitigation, and transformation. This contributes to a continual stream of validated upstream research products suitable for downstream adoption and adaptation. It also allows for a greater appreciation of the relevance of different disciplines' contribution to climate change research and action.

### 2.3. Upstream Research

Improved plant resilience for yield under various climate regimes is a critical component of climate research. Advances have been made in basic upstream science that enable researchers to understand better the roles and relationships of molecules that make up cells of any organism, and to understand better plants and their performance. Omics refers to the collective technologies used to explore these molecules and enhance our understanding of plants and their performance. This understanding enhances the opportunities to design solutions for organismal responses to altered biophysical conditions such as drought, heat, and submergence in rice and other crops.

Two considerations are important in dealing with molecular data. First, the obvious one that a molecule does not function in isolation and there is much value in identifying and understanding its network interactors. For example, a metabolite, such as a hormone, can affect various traits through

its interaction networks [28]. The case for understanding interactors was recently highlighted by the discovery of a multi-gene quantitative trait locus (QTL) for rice yield under drought [29]. Further interactors with genes in this QTL, and at the genome level, were identified through omics studies [30].

The second consideration is that the functionality of a molecule may not be limited to a single function. Despite compelling cases of a single gene, protein or metabolite having a single function, multi-functionality of these biomolecules is becoming increasingly clear. For example, a gene can be spliced in variant forms of RNAs and spatio-temporally. Conditionally guided alternative splicing of the genes is now well known [31]. After translation into a protein, the same protein could function as a structural protein and/or as an enzyme [32].

What is required is a concerted effort to elucidate the possible alternative functions. Just as good bioinformatics analyses of a gene can predict its alternative splice variants, the next level of in silico, in vitro and in vivo research is required to predict the various potential functions of a protein or a metabolite. Importantly, a preponderance of such data from microbiology should be considered and queried for higher organisms. Omics studies, down to ionomics, in mutant microbes can be a good starting point to look at the effect of changes in a single gene/protein/metabolite/ion, etc.

Cell biology brings together multiple disciplines to advance its frontiers. The field that brings biology and microelectromechanical systems together is known as Biological MEMS (BioMEMS). It combines nanotechnology, mechanics, and cell biology. The field is already providing unique opportunities to study cell functions [33]. In effect, the history of scientific research has led us to a point of modeling life. From here, innovations in scaling upstream technologies are as important in understanding plant life and function towards food security as innovations in scaling the downstream products of agricultural research such as CSA.

## 2.4. Crop improvement

Drought-, saline-, and submergence-tolerant rice varieties are fundamental when it comes to climate risk management in rice-based systems in Asia. Plant breeding is the exercise of manipulating evolutionary biology to benefit agricultural systems. Central to the success of a breeding program is the generation of genetic variation and the evaluation of the resulting progeny in the environment of interest. Resource constraints on plant breeding programs demand that evaluation of selection candidates occurs only at a number of locations, which collectively serve as a representative sample of what is in reality a targeted population of environments [34–36]. Within this sample of environments, the ranking of breeding material results in selection decisions that move allele frequencies within breeding populations towards enhanced adaptation to the larger targeted population of environments.

Change in allele frequencies over time, for quantitatively inherited traits, is the definition of evolution, and is one of the primary mechanisms that natural populations survive amid changing environments [37–39]. However, the speed with which evolutionary change drives change in allele frequencies is directly proportional to the generation interval of the population under selection. In other words, if climate change occurs quickly and a natural population reproduces too slowly, the species may soon face extinction.

Breeding populations are subject to the same laws of quantitative inheritance as natural populations; however, to date, most of the effort to use plant breeding to combat climate change has focused on the creation of climate-ready varieties through classical molecular genetic approaches. This strategy usually involves the identification of a particular stress which is predicted to increase under current climate models (such as heat stress or drought stress) followed by a search of exotic germplasm for specific large effect genes capable of forming a stress-tolerant variety when backcrossed into a modern cultivar. Although this strategy has met with some success, as evidenced by the identification of stress tolerance genes in rice [40], these quantitative trait loci (QTL) generally do not fully explain the phenotypic variance for a trait, are usually rare, and are predicated on the assumption that the stresses under which the QTL was identified mirror the anticipated stresses imposed by climate change.

To better leverage the power of quantitative genetics to ensure agricultural systems are robust to climate change, the scientific focus should expand beyond creating climate-ready varieties to include the design and the development of climate-ready breeding programs. Such a program is characterized by a rapid-cycle breeding strategy that imposes accurate selection on elite (but genetically variable) breeding populations as the climate changes [41]. Several key developments in breeding technologies in recent years have enabled this transformation to take place. Genomic selection, for example, is a strategy by which a DNA fingerprint can be used to borrow information from related lines to predict the value of a new untested breeding line [42].

Taken together, and combined with a well-designed multi-location testing strategy, rapid recycling of breeding lines based on predicted performance of new material can impose selection on all genes at the same time by shifting their allele frequencies in ways that favor adaptation to the current climate. Rapid cycle recurrent selection in the most recent climate, as it changes, has the potential to fully leverage the natural ability of crop species to adapt to climate change. Climate-ready breeding programs are capable of delivering a steady stream of improved varieties to farmer's fields at the pace of climate change. This is because the last 3–5 years of breeding trial data are always informing and updating the selection of new breeding lines among thousands of candidates, ensuring that seed systems have access to well adapted and high performing new varieties.

This system already exists among commercial breeding programs serving areas characterized by industrial agriculture. However, across the developing world, national agricultural research and extension organizations (NAREs) operate in relative isolation to one another and evaluate, at best, a few hundred new lines annually at relatively few testing locations and observe the performance of lines for many years before re-use as a parent to complete a full breeding cycle [43]. This has kept both selection intensity (number of new lines created) and selection accuracy (number of test sites) low, while increasing the number of years required to complete a breeding cycle.

Enabling positive change in research for development towards climate-ready breeding programs will require strategic partnerships between advanced research institutes (ARIs) and NAREs. These networks must enable distributed testing across a global region, develop and transfer economical genotyping technology to enable genomic selection, and enable the proper analysis of genotype and phenotype data to make accurate predictions. They must also encourage the rapid recycling of parental material to ensure that the most recently adapted genetics are recycled as fast as the climate they are tested in is changing.

*2.5. Nutrient Management*

Soils are a fundamental component of food crop production ecosystems, including rice-based systems. Soil management is critical to productivity levels and to sustainability of production systems over time. Lowland soils differ from upland soils, and often rice can be grown in these lowlands soils without the addition of fertilizer due to existing biological nitrogen fixation and enhanced availability of other soil nutrients, however, phosphorous yields remain low. As a result, nutrient management is considered critical for sustainable rice production and the attainment of high yields especially in the context of climate change. Overuse of fertilizers pollutes water sources and generates nitrous oxide, a powerful greenhouse gas [44].

The authors of [45], in Japan, and the authors of [46], in China, observed increased rice yields under high atmospheric concentrations of carbon dioxide when nitrogen fertilizer was applied in large amounts. Similarly, a meta-analysis to evaluate two scenarios of climate change, i.e., elevated concentrations of carbon dioxide and elevated ozone concentration, showed increased rice yield by 12% with elevated carbon dioxide [47]. However, yields were lower when elevated carbon dioxide was imposed with low levels of nitrogen. These results demonstrate the need for nitrogen fertilization for rice production under climate change scenarios of increased atmospheric carbon dioxide.

The site-specific nutrient management (SSNM) approach developed in the 1990s to calculate field specific requirements for fertilizer nitrogen, phosphorus, and potassium for cereal crops based on

scientific principles [48], offers climate change adaptation potential for rice. SSNM improved rice yields versus farmer practice, which is often based on blanket recommendations [48], while reducing fertilizer application in some situations [49]. The increase in grain yield while lower amounts of fertilizer are applied has been associated with increased timing of application, particularly for nitrogen, which aims to match critical demand of nitrogen at different growth stages. This enhances nutrient use efficiency, while reducing losses to the environment, including greenhouse gas emissions.

SSNM, therefore, serves as a climate smart technology that enhances resource use efficiency while reducing greenhouse gas emissions. It can be used to adjust nutrient management options for different climate and local condition scenarios and potentially contribute to sustainable rice production. The dissemination of SSNM to smallholder farmers can be achieved at scale using Information and Communications Technology (ICT) decision support tools such as Rice Crop Manager (http://cropmanager.irri.org).

### 2.6. Water Management and Greenhouse Gas Production

Water management is a critical component of both climate change adaptation and mitigation strategies. A range of technologies is available that can increase water productivity [50] by reducing irrigation input to rice fields without reducing the yield: optimum irrigation scheduling [51], irrigation method [52], field design, and land leveling. Strategies to manage water demand include a change in the cropping calendar, choice of crops, and cultivation practices [53]. One very promising approach is alternate wetting and drying (AWD).

AWD was developed by IRRI and its partners in the early 2000s as a water-saving technology for drought-prone areas [54]. However, the practice in which rice fields are not kept flooded continuously but exposed to a number of dry periods throughout the growing season also has a high methane mitigation potential [55]. Methane is a greenhouse gas, and by oxygenating the soil, methanogenic bacteria are inhibited and methane is oxidized, this reduces methane emissions by 14–80% (mean 43%) [56].

The AWD technology has been integrated in various national recommendations for good rice crop management. Driven by international discussions on greenhouse gas mitigation, particularly after the Paris Agreement and the development of Nationally Determined Contributions (NDCs), AWD has been increasingly seen as mitigation technology and gained new attention. Several rice-producing countries mention AWD directly in national plans for greenhouse gas emission reduction (e.g., Vietnam and Bangladesh in their in NDCs and the Philippines in the AMIA program, which is program of the Department of Agriculture), others refer to water-saving techniques such as AWD as a mitigation measure (e.g., Indonesia and China in their NDCs) and design strategies for large-scale dissemination of the technology.

### 2.7. Understanding Synergies and Trade-off of Solutions to Climate Change

The urgency of climate change is driving researchers and practitioners to come up with solutions to minimize the impact of climate change across different cropping systems. As listed in Section 2.1 Section 2.2 Section 2.3 Section 2.4 Section 2.5 Section 2.6, so often the model of employing a modern/improved technology is presented as a potential solution; however, most of these technologies are evaluated based on short term gains in the resiliency of food systems. Although the positive effect/synergies of these technologies are well evaluated, there is little attention given to trade-offs. Many of these technologies are intrinsically linked with other ecosystem functions. The gain by intensively achieving one target may limit or degrade other targets. Although DSR, nutrient management, and AWD provide many benefits in terms of climate change adaptation and mitigation, they also have many potential trade-offs. Balancing methane versus nitrous oxide emission is one example of these trade-offs [55,57].

Similarly, paddy fields are the largest human-made wetland occupying ~18% of total global wetland area [58]. Along with food, the diverse multi-functionalities of rice paddy fields, including

rich biodiversity, make it essential for a wise use of these wetlands. The concept of some of these technologies, like dry seeded rice and AWD, is already a couple of decades old, but these technologies have not been widely adopted in paddy ecosystems. Was it because of lack of adoption drivers or other secondary problems which arise due to adoption of these solutions? This is another example of the importance of an interdisciplinary approach to understanding synergies and trade-offs of potential technologies that could minimize the climate change impact.

## 3. Trans-Disciplinary Networks for Climate Change Responses

### 3.1. Perceptions of Risk and Social Equity: Broadening the Research Networks

Participatory and collaborative research brings different stakeholders together to identify common challenges, and build structural and cognitive social capital in the process. This is often new territory for researchers working in agricultural science.

Social differentiation may imply varied vulnerability and capacity to adapt to climate change, climate variability, and other stressors. Although some social norms and relations seem fixed, others are fluid and flexible, especially in times of social change. How resources are accessed, distributed, and consumed, and how labor is divided into productive and reproductive tasks affect how farmers perceive risk, prioritize and share tasks in everyday farming, experience hardship, and shape aspirations about future livelihoods. All this will influence the adaptation space. To be successful, climate change adaptation and technology adoption must take these conditions into account [59].

Perception of climate change has been studied in many high income countries with diverse populations. Often, however, these countries are not as critically affected by climate change as many low income countries are. It has been argued that climate change is psychologically distant for people in high-income countries [60,61]. It is suggested that reducing peoples' psychological distance to climate change and highlighting its proximal consequences will increase sustainable behaviors [61]. Farmers' perceptions of climate change risks are an important factor influencing their adoption and adaptation strategies. The way, individuals interpret their own risks, and societal risks affect what kind of adaptive behavior they are likely to take [62]. Perception and acceptance of risk in general and climate change risks have been shown to have their roots in social and cultural factors [63].

Farmers in many low-income countries are directly affected by climate change, and their livelihoods are threatened due to adverse weather conditions impacting on crop productivity. It has been shown that farmers in Pakistan perceive various climate risks including extreme temperatures, animal and human disease, crop pest, and droughts [64]. Farmers' sensitivity to climate change depends on the availability of resources, and their adaptation to climate risks is subject to various constraints [64]. Farmers, who believe climate change is happening and influencing their family's lives, perceive higher risks than farmers who do not think so [62]. This has also been shown in high-income countries where farmers' pro-environmental behavior is limited by conceptual, practical, and informational barriers [65].

There are gender-specific information and capacity needs that are critical for adoption of CSA [27]. Access to knowledge and information is limited for poorer women in rural areas. Extension and advisory services have historically been unsuccessful in reaching women in general [26]. This can be attributed to social and cultural norms where men in the household are considered "farmers", whereas women are only considered "helpers" on family farms. Extension services therefore generally reach out to men as heads of households and farmers and breadwinners. There are also fewer formal social networks for many groups of women compared to men. This again limits women's access to information and knowledge. It would be naïve to rely completely on informal farmer-to-farmer knowledge exchange to make information accessible to all categories of women.

A deep understanding of preferred information channels and trust in those channels are important. Although mobile phones are being lauded as a panacea for this, there is evidence of a large gender digital divide and the challenges associated with it [66]. Studies have also shown that socially marginal

castes in south Asia have less access to extension advice and, therefore, are at a disadvantage when it comes to accessing information and technologies [24]. What is important in this context is the capacity for development of rural extension and advisory services to be gender-responsive and effectively reach out to female client groups.

It is necessary to understand farmers' realities in order to propose policies that will make their farming system more climate resilient [67,68]. The choice of climate smart technologies depends on the livelihood portfolio of rural households and an in-depth analysis is required to suggest options. For example, smallholder or landless households in marginal areas often depend on livestock for their livelihoods. On the face of it, livestock systems are professed to be undesirable, but if policies chose to curb those systems (as part of mitigation efforts), scores of households would be bereft of income, food and/or nutrition sources. Similarly, it makes little sense to have policies to promote labor-intensive CSA in areas where there are labor shortages because of migration.

Intra-household and intra-community trade-offs related to use of climate smart technologies and approaches with respect to gendered roles and responsibilities and associated labor and benefits need to be thoroughly assessed. Unpacking intra-household variation in perception of risks and choices/use of coping strategies is critical. This is again the result of a mix of roles and responsibilities, social norms, risk perceptions, and access to resources. Research and policy must disentangle social processes and practices and be sensitive to intersecting inequalities that emerge when climate change impacts and responses cut across age, class, ethnicity, gender, and space.

### 3.2. Linking Researchers, Practitioners and Policy-Makers for Scaling of CSA

The role of technology development is fundamental to climate risk management. The aforementioned technologies and innovative practices represent a fraction of those that fall within the framework of CSA. Meeting the complex and urgent challenges presented by climate change requires moving beyond interdisciplinary research to transdisciplinary and cross-sectoral cooperation to integrate knowledge from diverse stakeholders committed to tackling complex social and ecological problems. For example, farmers themselves, along with NGOs and others who have continuous and long-term engagement in communities, should be included during the process of knowledge co-production and for co-designing appropriate climate change mitigation and adaptation strategies that are ethical and meet the needs of the target group. These needs may include the demand for an early maturing crop variety that is better suited to a changing planting period or for improved management and post-harvest technologies, but they may also include shifts out of agriculture into other livelihood opportunities.

Teams that take into account the complexity of social and ecological interactions will be better able to identify appropriate solutions. Such teams can tap into the upstream development of stress-tolerant varieties that may be appropriate and also provide valuable information to breeders regarding user preferences. This contributes to the development of suitable crop varieties, the cultivation of which is enhanced by complimentary land and management as well as harvest and post-harvest management practices.

However, technology development per se is not enough. There are many examples where technologies and practices, while technically very sound, have not scaled. In the case of AWD, for example, farmer adoption in some countries has been slow. In the Philippines, ~80,000 farmers have adopted AWD on ~90,000 ha (~6% of total irrigated rice area). In Vietnam, on the other hand, there has been large-scale adoption of AWD in major rice production provinces in the Mekong River Delta—Dong Thap and An Giang—according to a recent study based on satellite data, household surveys, and in situ moisture readings [69].

There are many reasons for different rates of farmer adoption of CSA, including AWD, and these have been well documented [25,70,71]. What is clear from these studies is that successful scaling of CSA is dependent on institutional and organizational capacity, along with government support

and infrastructure development. This requires researchers to foster networks with practitioners and policy-makers.

IRRI supports the process of AWD dissemination by providing evidence for the technology's benefits, mapping tools to assist planning and investment opportunities, and strengthening the capacity of extension services. In the Philippines, IRRI conducted a climatic suitability analysis for AWD on a national scale [72]. A similar, but more in-depth, assessment was conducted for Vietnam with a stronger focus on high priority provinces for mitigation action in order to guide the implementation of AWD. The Vietnam case is a good example of trans-disciplinary research accomplished by a combination of high-resolution GIS assessment, socio-economic feasibility analysis, and strategic contributions from local stakeholders. Since 2016, IRRI has provided information and supported training of a local network of NGOs, government organizations, and civil society organizations in Bangladesh—the Northwest Focal Area Network (FAN)—in Rangpur and Rajshahi divisions. The network partners used their individual training models to disseminate the AWD technology to farmers in their respective area of authority.

Similarly, scaling of mechanization and post-harvest technologies and practices has been challenging. Post-harvest losses from harvesting to milling can reach 14–40% compared to 6–8% when using the best practice management options available. The introduction of dryers that were technically and economically feasible, for example, has failed in most cases and an estimated 80% of the rice in Southeast Asia is still sun-dried. This is because projects usually focused on the technology and did not facilitate market access for a better quality paddy produced at higher cost, whereas existing traders and millers were not interested in purchasing premium products. Scaling agricultural machinery requires identification of appropriate machinery along with production, dissemination, servicing and financing.

The establishment of networks provides institutional and organizational context for longer-term engagement with key stakeholders, an engagement that goes well beyond the normal 3–5 year lifetime of a research project. The authors of [73], for example, documented the long-term evolution and successful adoption of the rice flatbed dryer in Vietnam through continuous involvement of a research and development team from Nong Lam University. The impact of their work came as a result of sustained interaction with a tight network of partners, working in the same innovation trajectory, for 25 years. In the process they developed major improvements to the original design and a new type of dryer emerged. Similarly, laser-assisted land leveling, which can lead to productivity gains, water saving, and reduced energy use and emissions from rice [74], took more than six years to evolve from the introduction of the first demonstration unit to the emergence of a service economy around laser land leveling which resulted in a significant increase in sales. It took more than 10 years before there were ~10,000 units being used by service providers covering around half a million ha in India [75].

*3.3. Trans-Disciplinary Networks for Climate Change Transformation*

There are growing examples of trans-disciplinary networks for climate change transformation. One such example is the Direct Seeded Rice Consortium (DSRC), a public–private multi-stakeholder research-for-development platform, established by IRRI, to address complex research issues and scaling direct-seeding of rice (DSR) in Asia. DSRC aims to improve environmental and economic sustainability of rice production by developing and scaling science-based comprehensive mechanized and precise DSR practices through public–private partnerships. DSRC brings together public and private sector partners, including researchers, from across South and Southeast Asia, as well as Advanced Research Institutes such as Cornell University and the University of Sydney. DSR offers both adaptation (adapt to water shortage and weak and variable monsoon) and mitigation (reduction in GHG emissions) options to climate change [53,76,77]. Data and results-sharing through IRRI's Open Access and Data Management Policy, together with Research Data Management best practices, are available to all consortium members and this allows for dialogue and sharing of experiences.

DSR has emerged as an efficient, economically viable and environmentally promising alternative to Asia's most dominant method of rice production known as puddled transplanted rice (PTR) as it

addresses the major drivers of rural change in the region, especially rising labor and water scarcity. In DSR, unlike PTR in which rice seedlings are first established in a nursery and then transplanted into a puddled main field, rice is directly sown in the main field. This eliminates the step of nursery raising either by using dry seeds in non-puddled soil (dry seeding- 'Dry-DSR') or pre-germinated seeds in puddled soil (wet seeding- 'Wet-DSR') [76]. DSR reduces the cost of cultivation and GHG emissions, and increases farmer's income without yield penalty [53,78]. Based on a meta-analysis, DSR has reduced methane emissions from 40 to 63% compared to PTR especially dry-DSR, because of less flooding and more aerobic conditions which prevents methanogenesis and therefore methane emissions. However, the more aerobic environment in DSR also creates conditions for $N_2O$ emissions. With better nitrogen and water management, there is further scope to reduce both $CH_4$ and $N_2O$ emission in DSR.

There are few studies comparing DSR and PTR in terms of global warming potential (GWP) taking into account both $CH_4$ and $N_2O$. Generally, GWP has been found to be lower in DSR in China [79], India [78,80], and in Japan [81,82]. Under elevated $CO_2$ condition, emission of $CH_4$ in the paddy field may, however, increase [83]. Therefore, it is important to develop, refine, and deploy alternate low emission rice establishment methods such as DSR so that yield-scaled GWP of rice production is reduced.

DSR adoption in Asia has nonetheless been low because of some risks/constraints associated with the practice, and poor market development of products critical for DSR success in new areas. To overcome some of these research and market gaps and catalyze wide-scale adoption of DSR, a trans-disciplinary approach, characterized by public–private partnership, is needed. Networks like DSRC enable multi-sectoral collaboration and create synergies among various stakeholders. They also provide a platform for exchange of knowledge, ideas, and technologies across actors and countries, and therefore generate impact much faster.

Networks are not just critical for the development and scaling of CSA, they also drive the transformative change needed to address climate change. Threats of sea level rise demand trans-disciplinary networks applying landscape- and systems-thinking given that damage will not be limited to specific production systems. The dense populations of the Asian mega-deltas rely on intensive land use systems, namely, rice production and aquaculture. Proposals to tackle the threats of climate change in Asian mega-deltas include building a network across research centers that specialize in fisheries/aquaculture, rice production, and water management to tap into the diverse knowledge systems and long-standing in-country experiences across the region.

Poverty in many vulnerable areas of the world and uncertainty about the future impacts of climate change stymie comprehensive adaptation and planning. Currently, relatively little is being done to anticipate or prepare for the potentially devastating impact of sea level rise [84]. In the case of climate change, decision-makers have generally avoided taking action on the premise that uncertainties make decisions difficult [85]. Uncertainties are not specific to climate science. In other domains, such as finance, uncertainties have not prevented humankind from creating methods to reduce uncertainty and plan according to multiple scenarios. Indeed, there is wide scientific consensus and plenty of data to support climate trends, and these can be used to model future climate impacts and plan in the face of uncertainty.

We argue that this attitude should also prevail with respect to sea level rise, especially for countries in Asia that host mega deltas. Although scientists and policy-makers are acutely aware of the mounting dangers of climate change, they continue to delay making decisions and avoid taking appropriate action. At the very least we must focus on removing current known barriers to future planning for farmers, including access to credit, savings, timely inputs, land titles, and reducing gender inequity, that will empower farmers to make adaptation decisions. An interesting example of a transdisciplinary approach to mega-deltas and sea level rise is the Living Deltas Research Hub that is coordinated by Newcastle University in the United Kingdom. The hub operates across four mega-deltas including the Mekong in Vietnam and the Ganges–Brahmaputra–Meghna system in Bangladesh and India. The Hub

epitomizes South–South and North–South partnerships and brings together academia, business, NGOs, government, and local communities to strengthen people's livelihoods in the face of challenges such as sea level rise (https://www.livingdeltas.org/).

Migration and land use change also contribute to the transforming landscape which will accelerate under climate change. This will require out-of-the-box thinking for food system networks to expand in unconventional ways to include health, emergency response, and environmental protection professionals. Climate change will alter the fine balance of spatial and temporal windows for agricultural production leaving many households with no other opportunities but to migrate to cities since other income opportunities in rural areas are limited. However, populations are growing in many developing countries, and, together with a shift away from food production to urban living, food insecurity is likely to be exacerbated by other drivers of migration, e.g., labor demand in cities and the changing lifestyle preferences of youth.

Claims about migration driven by climate change have received mixed responses ranging from general agreement to complete rejection. This is a futile discussion given that vulnerabilities of rice farming systems and communities need to be assessed through a variety of trans-disciplinary concepts including sensitivity or susceptibility to harm and lack of capacity to cope and adapt to changing conditions. Small-scale farmers deal with constantly changing conditions as it is, regardless of whether these are attributed to climate change [86]. Overwhelmingly, they currently lack the ability to cope with destructive events [87]. It is imperative to reduce uncertainties for policy-makers which will enable them to plan for a stable future. There is a need to identify hot spot areas of vulnerability where there are measures available for increasing resilience of rice farming systems by introducing (i) rice varieties or alternative production options (i.e., shrimp/fish) that are tolerant to salinity and flooding stresses, (ii) improved crop/water management, and (iii) real-time forecasting of salinity and flood threats to adjust cropping calendars and management practices to better plan and cope.

## 4. Conclusions

Key to agricultural climate change adaptation, mitigation, and transformation is fostering trans-disciplinary networks. This paper highlights progress on developing these networks in the context of rice-based farming systems in South and Southeast Asia. Trans-disciplinary and cooperative efforts within the context of innovation systems are needed to increase farmers' access to and use of climate smart technologies and practices. A broad trans-disciplinary treatment of the problem of climate change and climate variability in agriculture through the application of CSA will transcend disciplinary demarcations facilitating insights into the problems limiting adoption at scale, while helping to identify possible novel solutions.

Farmers' realities are so diverse that agricultural innovation requires assistance from a variety of disciplines working together. This represents a challenge to researchers who are traditionally channeled by disciplinary training into narrow specialisms. This contrasts with many farmers who not only manage and experience the whole of their farming system, but are also increasingly working off-farm and earning more of their income from non-farm sources [88]. Researchers are part of an impact pathway, and they are increasingly working together in interdisciplinary teams in the pursuit of agricultural innovations that meet farmers' needs. The challenge posed by climate change, however, is very complex and can only be met by a trans-disciplinary response, ones that bring together researchers, practitioners, and policy-makers. Responses to climate change challenges in rice-based systems in South and Southeast Asia illustrate the type of interdisciplinary research that is required globally and the types of trans-disciplinary networks needed to further climate change adaptation, mitigation, and transformation, and ensure that CSA makes substantial contributions to the realization of the Sustainable Development Goals.

**Author Contributions:** Conceptualization, J.H., J.B., E.F., M.C., T.K., and S.Y.; writing—original draft preparation, J.H., J.B., E.F., A.K., M.C., S.Y., V.K., T.J.K., B.O.S., J.C., K.N., T.S., R.P., P.C., and M.G; writing—review and editing, J.H. All authors have read and agreed to the published version of the manuscript.

**Funding:** This work was supported by the CGIAR Research Program (CRP) on Rice Agri-food Systems (RICE, 2017-2022).

**Acknowledgments:** The authors would also like to thank three anonymous reviewers who provided invaluable comments on an earlier version of this paper.

**Conflicts of Interest:** The authors declare no conflicts of interest. The funder had no role in the design of the study; in the collection, analyses, or interpretation of data; in the writing of the manuscript; or in the decision to publish.

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
