# Peer review of "Trans-Disciplinary Responses to Climate Change: Lessons from Rice-Based Systems in Asia"

_climate, doi:10.3390/cli8020035_

Round 1

Reviewer 1 Report

The aim of this submission is to illustrate ways to foster trans-disciplinary research on climate change and to understand how South-North and South-South partnerships can be enhanced for building climate resilience. This is an excellent submission that is well written and important for framing the often discipline fragmented responses to the climate challenge especially for agriculture.

The first part of the paper on approaches to build resilience is well presented although the potential role of crop modelling and real time remote sensing assessment is missing from the presented approaches. Authors should however strengthened the part on building South-South and South-North partnerships as I feel this has not be well documented how the partnerships worked and in what ways they improved or made solution development easy or more complex. This is the part that can be further strengthened in the submission.

Line 23-34: The abstract should already highlight some of the networks and the sectors that they have been used. Line 43: Geographical location alone cannot explain vulnerability. What aspects of the location makes the vulnerable, is it heavy reliance on natural systems? Line 109: what explains the increase in discharge, explain. Line 187 to 221: The section can be strengthened by more references especially after line 202. Line 442: Also add the % for Bangladesh for understanding.

Author Response

The authors would like to thank the reviewer for his/her very useful comments and suggestions. We have amended the text as follows:

Reviewer - Authors should however strengthened the part on building South-South and South-North partnerships as I feel this has not be well documented how the partnerships worked and in what ways they improved or made solution development easy or more complex. This is the part that can be further strengthened in the submission.

Authors' response - We have added text, please see lines 522-528 of the revised version.

Reviewer - Line 23-34: The abstract should already highlight some of the networks and the sectors that they have been used.

Authors' response - We have referred to the Direct Seeded Rice Consortium (DSRC) in the abstract

Reviewer - Line 43: Geographical location alone cannot explain vulnerability. What aspects of the location makes the vulnerable, is it heavy reliance on natural systems?

Authors' response - We have amended the text to refer to the fact that smallholder farmers often farm marginal land

Reviewer - Line 109: what explains the increase in discharge, explain.

Authors' response - we have added that this was due to increased precipitation in the watershed

Reviewer - Line 187 to 221: The section can be strengthened by more references especially after line 202.

Authors' response - We have amended the text and added the following references:

Chen, L.; Tovar-Corona, J.M.; Urrutia, A.O. Alternative Splicing: A Potential Source of Functional Innovation in the Eukaryotic Genome. Int. J. Evol. Biol. 2012, 2012, 1–10

Tawfik, O.K. and D.S. Enzyme Promiscuity: A Mechanistic and Evolutionary Perspective. Annu. Rev. Biochem. 2010, 79, 471–505.

Sanati Nezhad, A. Microfluidic platforms for plant cells studies. Lab Chip 2014, 14, 3262–3274.

Reviewer - Line 442: Also add the % for Bangladesh for understanding. 

Authors' response - We removed the reference to Bangladesh so that reference is made to the Philippines where we have the percentage figures.

Reviewer 2 Report

The work can be published without further modification.
The citation  on line 38 must be inserted in the text as a number, as well as the quote on line 485 (Gummert et al.) which, however, is not found in bibliography.

Author Response

We are grateful to the reviewer for his/her endorsement of the manuscript.

We have inserted the citation on line 38 as a number. We have, likewise done so for the citation on line 485 (Gummert et al.) and added this to the bibliography.

All changes are in track changes in the revised submission.

Reviewer 3 Report

See comments on the manuscript.

Author Response

The authors thank the reviewer for his/her suggestions on the annotated PDF version. 

We have revised the manuscript as per the reviewer's suggestions. All the changes can be seen in track changes and can be readily compared to the reviewer's annotated PDF version.

Round 2

Reviewer 3 Report

Thank you for considering some of my previous comments and edits. I find this version of the manuscript reads better. There are a few further comments and edits that should be checked in this second version.

Researchers are people too so the authors might consider making researchers part of the system, not identifying them as actors or players? There are some spelling issues that should be corrected.

Author Response

As per the reviewer's comments/suggestions, I have gone through the manuscript in detail and I've corrected for the missed spelling errors and corrected for where there were some remaining grammatical mistakes.

Again, my co-authors and I are grateful to the reviewer for his/her diligence.